# Efficient neural codes naturally emerge through gradient descent learning

Ari S. Benjamin [1] ✉, Ling-Qi Zhang [2], Cheng Qiu[2], Alan A. Stocker [2] & Konrad P. Kording[1,3]

Human sensory systems are more sensitive to common features in the environment than uncommon features. For example, small deviations from the more frequently encountered horizontal orientations can be more easily detected than small deviations from the less frequent diagonal ones. Here we find that artificial neural networks trained to recognize objects also have patterns of sensitivity that match the statistics of features in images. To interpret these findings, we show mathematically that learning with gradient descent in neural networks preferentially creates representations that are more sensitive to common features, a hallmark of efficient coding. This effect occurs in systems with otherwise unconstrained coding resources, and additionally when learning towards both supervised and unsupervised objectives. This result demonstrates that efficient codes can naturally emerge from gradient-like learning.

Careful psychophysical studies of perception have revealed that neural representations do not encode all aspects of stimuli with equal sensitivity[1]. The ability to detect a small change in a stimulus, for instance, depends systematically on stimulus value. A classic example of this is the so-called 'oblique effect' in which changes in visual orientation are easier to detect near vertical or horizontal than oblique orientations[2]. The fact that these sensitivity patterns are ubiquitous and widely shared between animals motivates us to study the potential underlying reasons why they exist.

The efficient coding hypothesis has become a standard explanation for the emergence of these non-homogeneous sensitivity patterns[3]. It predicts that sensory systems should preferentially encode more common aspects of the world at the expense of less common aspects, as this is the most efficient way (in the information-theoretical sense) to make use of limited coding resources. Indeed, perceptual sensitivity typically reflects the statistics of the visual environment[4–9]. While much is known about efficient neural codes and their link to the stimulus statistics and perceptual behavior, the mechanisms that give rise to such codes remain unknown.

Brains are not born with fully developed sensory representations. Many developmental studies of perception in infants and young children have shown that visual sensitivities improve with age and visual experience even until adolescence[10–13]. Much of the improvement depends on visual experience and is due to neural changes downstream of the retina[14–18]. This suggests that perceptual sensitivity depends on the neural representation of sensory information and how these representations change with experience during development.

We hypothesize that general, task-oriented learning rules provide a sufficient mechanism to produce efficient sensory representations in the brain. Notably, we hypothesize this is separate from and does not depend upon explicit efficient coding objectives or explicit coding constraints like noise. Driving our hypothesis is the idea that any gradual learning process can only learn so much at a time. An effective learning algorithm should thus prioritize learning more important aspects before less important ones. Conveniently, features that are more common are also easier to learn from a limited exposure to the world, in a learning-theoretic sense. These ideas are broadly equivalent to the notion that learning algorithms provide a second, implicit constraint on neural coding in addition to the explicit constraint imposed by the limited neural resources. Together, this inspiration from learning theory points to the possibility that effective learning algorithms will naturally produce better representations for common features, even if coding resources are otherwise unconstrained.

[1]Department of Bioengineering, University of Pennsylvania, Philadelphia, PA, USA. [2]Department of Psychology, University of Pennsylvania, Philadelphia, PA, USA. [3]Department of Neuroscience, University of Pennsylvania, Philadelphia, PA, USA. ✉e-mail: arisbenjamin@gmail.com

Our hypothesis directly predicts that we should find forms of efficient coding not just in biological neural networks but also in other learning systems. Especially, we expect that artificial neural networks trained to perform visual tasks to exhibit efficient neural representations similar to those found in the visual cortex despite their many differences in their local structural properties (e.g. noise) and connectivity. Since our hypothesis is general to the task, this should be observed for both supervised and unsupervised objectives, and should not require the explicit minimization of traditional efficient coding objectives like mutual information or reconstruction loss.

A study of the consequences of effective yet gradual learning on sensory representations must begin from a specific learning rule. One canonical learning rule is gradient descent, which proposes that neural updates improve behavior as much as possible for a given (very small) change in the overall weights. Though the brain may use more complicated learning rules, gradient descent is arguably the simplest rule that is effective for general learning and thus a baseline for theorizing about learning in the brain. If gradient descent produces efficient codes, this would provide a strong proof of principle that efficient codes can emerge from general-purpose learning algorithms.

To show that efficient coding emerges from gradient descent requires a formal understanding of how learning with gradient descent biases what is represented about the stimulus. This parallels an active effort in the study of deep learning. It is now recognized that what neural networks learn about their inputs is constrained implicitly by their learning algorithm, rather than by the architecture alone, as evidenced by their ability to memorize pure noise[19]. Many potential implicit constraints have been proposed to explain why large neural networks work well on unseen data (i.e. generalize)[20–23]. One prominent theory is gradient descent in a multilayer network supplies key biases about what is learned first[24–28]. This raises the possibility that such ideas could also demonstrate whether gradient descent learning is biased towards efficient codes.

In this paper, we describe how learning with gradient descent biases feature learning towards common input features, thus reproducing the relationship between stimulus statistics and perceptual sensitivity (Fig. 1). First, reproducing and extending previous results (see[29,30]), we show that deep artificial networks trained on natural image classification show similar patterns of sensitivity as humans. Then, to understand this effect, we mathematically describe how gradient descent causes learned representations to reflect the input statistics in linear systems. This effect occurs even in noiseless and overparameterized networks as well as for multiple learning objectives, including supervised objectives. To demonstrate that this framework can be applied to explain development, we also show that changes in sensitivity resembling changes in visual acuity in human children can be reproduced in a simple model trained with gradient descent on natural images. Our results show how learning dynamics provide a natural mechanism for the emergence of a non-uniform sensory sensitivity that matches input statistics.

## Results

Humans and animals show sensitivity that depends on the orientation of stimuli. In humans, the sensitivity of internal representations can be inferred from psychophysical data on discrimination thresholds[6] or from the empirical distribution of tuning curves in V1[18,31] (Fig. 2a). In many animals, internal representations are most sensitive at near vertical and horizontal orientations[2]. Since these orientations are also those that occur most often, this pattern of sensitivity can be seen as reflecting an efficient code for the natural world[7].

To ask if neural networks would show a similar phenomenon, we first obtained a set of relevant networks and measured their response to artificial stimuli. We chose to investigate deep neural networks trained on the ImageNet task[32] as such networks show a number of other similarities to human ventral stream visual processing[33–35]. We analyzed a range of architectures, including two large convolutional neural networks (CNNs), VGG16 and Resnet18, and Vision Transformers, which operate largely without convolution[36–38]. Then, to measure sensitivity, we measured the squared magnitude of the change in network activations given a change in the angle of oriented Gabor stimuli (Fig 2b; see Methods). For all three networks, we found that the internal representations were most sensitive to changes near cardinal orientations (Fig. 2c). These findings were robust to choices in the parameterization of Gabor stimuli (SI Fig. 3). The effect was more pronounced deeper in each network. The coarse pattern of sensitivity of ImageNet-trained deep networks to orientation is thus similar to that of animals and reflects the statistics of natural images.

We next investigated whether this pattern was due to factors inherent in the network or due to the statistics of the inputs on which it was trained. When set with random initial weights, the network architectures shown in Fig. 2 do not show this pattern, and shuffling the weights after training destroys the pattern (SI Fig. 1a). Changing the image statistics also changes sensitivity: when trained on a version of ImageNet in which all images are rotated by 45°, the networks lose sensitivity to cardinals and gain sensitivity to oblique angles (SI Fig. 1b). This finding recapitulates our preliminary findings and concurrent work of colleagues, and points to an origin in image statistics[29,30]. However, we also found that this effect is also partially learning-independent and due to architecture. Networks trained on rotated images do not simply rotate their pattern of sensitivity by 45° but instead partially retain increased sensitivity at cardinal orientations (SI Fig. 1b). In investigating the cause of this learning-independent component, we found that the use of spatial pooling with overlapping receptive fields (such as in AlexNet[39]) involves oversampling a rectangular grid and that this produces a significant cardinal sensitivity (SI Fig 1c). These analyses indicate that the pattern of orientation sensitivity is thus both a product of the input statistics and inherent factors like architecture.

To separate effects related to architecture and learning, we next examined the sensitivity of trained networks to changes in hue, as this is unlikely to be affected by rectangular convolutional processing. We

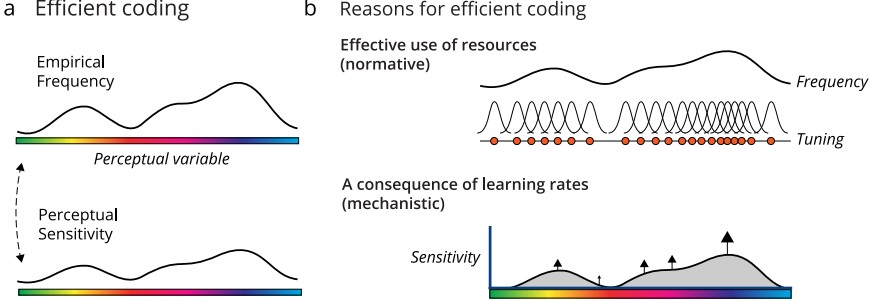

**Fig. 1 | Reasons for efficient coding. a** One consequence of efficient coding is that perceptual sensitivity reflects the empirical frequency of perceptual variables. **b** Efficient coding can be justified normatively as the most effective way to allocate finite neural resources to encode a stimulus ensemble. In this work we describe a mechanism for efficient coding due to learning components of the inputs at different rates dependent on their frequency.

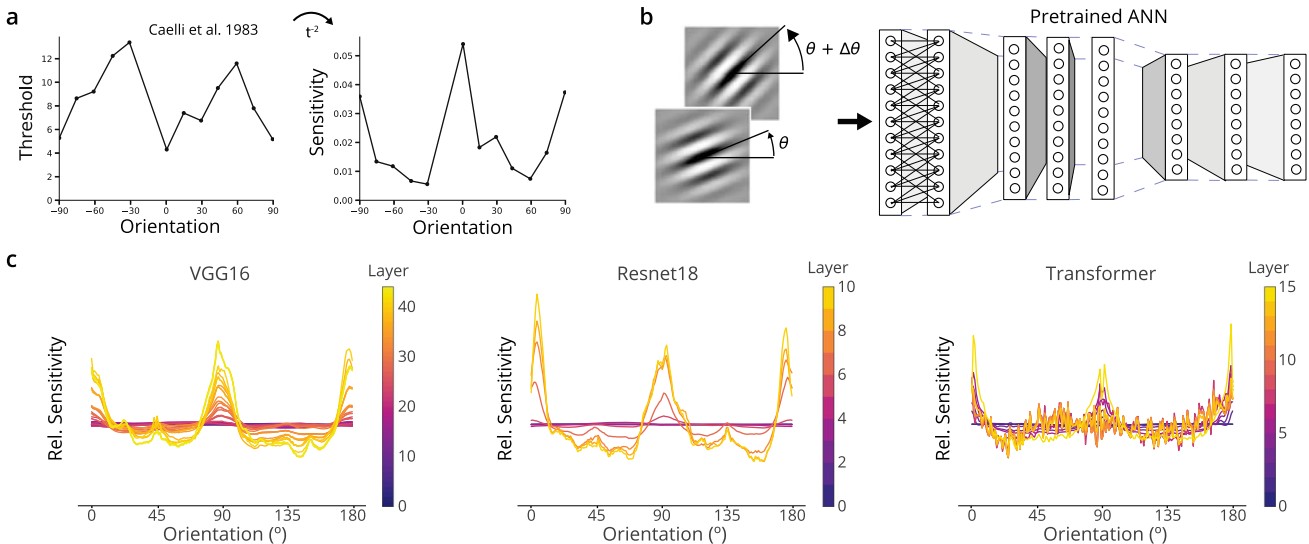

**Fig. 2 | Artificial neural networks trained to classify naturalistic images show similar patterns of sensitivity as humans. a** Discrimination thresholds for orientation vary systematically in humans. The sensitivity of the underlying internal representations, as the Fisher Information, can be inferred as the inverse square of the threshold[6,9]. Data from ref. [44]. **b** We measured the sensitivity of each layer in an artificial network as the change in layer's response due to a given change in orientation, i.e. the squared norm of the gradient with respect to orientation. **c** Relative (normalized) sensitivity to orientation for three networks trained on ImageNet, plotted for various layers in each network.

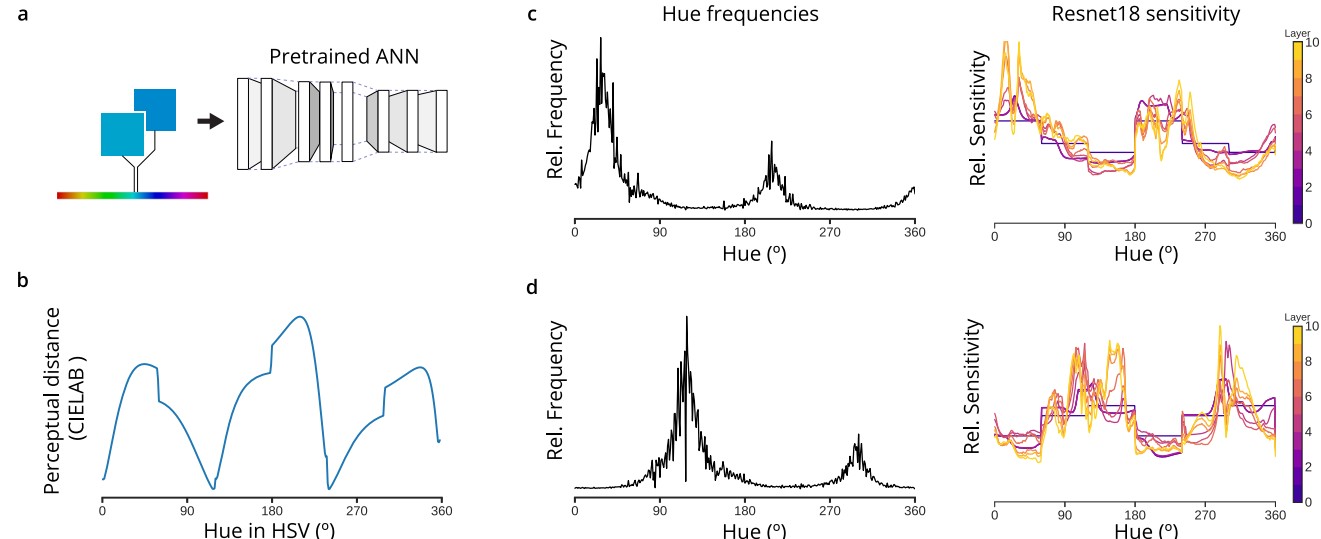

**Fig. 3 | The sensitivity of ANNs to hue also matches image statistics. a** The color of a uniform image is varied; in HSV color space, saturation and value are held fixed and the hue is varied. Results are averaged over possible saturations and values. **b** In humans, the sensitivity to the H axis can be inferred by the perceptual distance between uniformly spaced H values (calculated using the approximately perceptually uniform color space CIELAB) at S=V=1. **c** The sensitivity to hue in each layer in a trained ResNet18 tracks the empirical frequency of hues in the ImageNet dataset. **d** Training ResNet18 on a version of ImageNet in which hues are rotated results in a corresponding shift in hue sensitivity.

found that hue sensitivity was indeed related to the empirical frequency of hues in ImageNet (Fig. 3c) measured in HSV color space. Furthermore, the location of the peaks of network sensitivity roughly matched the peaks of human sensitivity to the hue axis of HSV color space (Fig. 3b). To test if this pattern is causally related to the input statistics, we trained a Resnet18 network on a version of ImageNet in which the hue of all pixels was shifted by 90˚. We observed a corresponding shift in the hue sensitivity (Fig. 3d). This suggests that in general the frequency of low-level visual features determines the sensitivity of artificial neural networks trained on object classification.

What is the origin of this phenomenon? In psychophysics, one leading explanation proposes that there is some constraint that limits the amount of information a neural population can contain about its inputs. These constraints may be noise, a finite number of neurons, or a penalty upon their activity. Due to this constraint, an optimal code will allocate more resources (and be more sensitive) to inputs that occur frequently[6,40]. However, deep networks can encode an extremely large amount of information; for example, they are capable of memorizing millions of examples of random noise[19]. Also note that these networks above are noiseless during evaluation and are overparameterized, in the sense that internal layers contain a greater number of nodes than there are input pixels. While typical networks do regularize their weights, this regularization is small and does not alone explain their ability to generalize to unseen examples[19]. These reasons suggest another effect may be at play besides inherent and unresolvable architectural constraints.

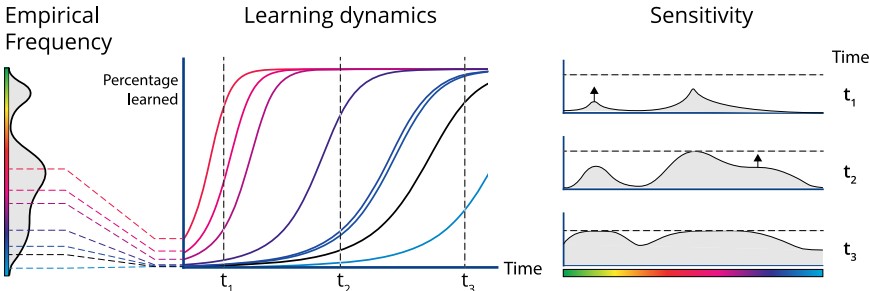

**Fig. 4 | Schematic of how the learning dynamics of linear networks causes a correspondence between network sensitivity and input statistics.** The learning problem is broken into components, each of which learns at a specific rate. The frequency or variance of a feature of the input data (e.g. the color red) in part determines the learning rate of the components that encode it. This means that the network becomes sensitive to frequent features first. Training may end before all features are fully encoded.

An alternative possibility is that incremental learning via gradient descent naturally leads to a frequency/sensitivity correspondence. This would allow this effect to occur in systems with otherwise unconstrained coding resources. This hypothesis relates to the idea from connectionist models of development that the most general aspects of a problem are often learned first[41]. To investigate this possibility, we analyzed a category of artificial neural networks amenable to mathematical study: deep linear networks. Deep linear networks contain no nonlinearities and are equivalent to sequential matrix multiplication. Despite their simplicity, deep linear networks show many of the same learning phenomena as nonlinear networks[24,28] and humans[42,43]. Moreover, this simplified setting allows us to separate the effects of gradient descent from those of network nonlinearity.

This linear setting allows us to characterize the dynamics of learning and the emergence of preferential sensitivity. At a high level, the network becomes responsive to features earlier when those features are more common (Fig. 4). When learning ends due to finite training time, finite data, or saturating performance, there is a residual higher sensitivity for common features. This can be seen mathematically, as we describe below and in two levels of increasing formality in the Methods and Supplementary Methods. Overall, the link between learning rate and input frequency, combined with finite training time, is an additional inductive bias beyond what features are useful for the task and means that trained networks will tend to be more sensitive to frequent features.

To concretely demonstrate this phenomenon we will focus on the task of reconstructing natural images with a linear network (Fig. 5). This simple unsupervised task is useful to guide intuition, but we emphasize that this theory also applies to supervised tasks (see below). This network can be as shallow as a single layer, in which case the reconstructed images are given by the matrix multiplication $\hat{X} = WX$. Importantly, this problem can be solved exactly with the solution that the weight matrix $W$ is the identity matrix $I$. This is thus an unconstrained encoding problem; if there is any non-uniformity in the sensitivity of the output $\hat{X}$ to changes in $X$ it must be due to the implicit constraints posed during learning. Analyzing the output sensitivity in this simple model will help to better understand the implicit preferences of learning with gradient descent.

In our demonstrative task we will examine the sensitivity of the output $\hat{X}$ to the magnitude of each principal component that makes up an image (as provided by PCA on the inputs, Fig. 5b) and describe why and how it tracks the input statistics. The mathematical analysis for this feature is much simpler than, say, for orientation. In this case also we have some expectation as to what pattern of sensitivity the efficient coding framework predicts because the principal components (PCs) are ordered by their variance. An efficient code in the presence of independent internal noise should be more sensitive to higher-variance PCs. Indeed, earlier PCs are composed of lower spatial frequencies, and humans are better at detecting changes in lower

spatial frequencies (Fig. 5e). If gradient descent provides a similar effect, we should find that the output $\hat{X}$ becomes sensitive to lower PCs first.

In a linear model it is possible to describe analytically how the sensitivity changes due to gradient descent. We first decompose the weights $W$ via singular value decomposition (SVD), $W = USV^T = \sum_i \sigma_i u_i v_i^T$, as a product of unit-length singular vectors ($u, v$) and their corresponding singular values $\sigma_i$. The evolution of these components under gradient descent is known as long as certain basic conditions are met, such as a very small weight initialization[24,28]. One key previous finding is that the singular vectors $v$ of the weight matrix rotate to align with the PCs of the inputs (see Theorem 2 in the Supplementary Methods)[24]. Due to this alignment, we find that the sensitivity of the output $\hat{X}$ to the $i$th PC is controlled by the size of the corresponding singular value in the weights, $\sigma_i$. This is more formally derived in Methods. For example, if $\sigma_i$ remains near its initialization close to zero, then the projection of data upon the $i$th PC will be filtered out and the output will not be sensitive to the corresponding PC. The sensitivity of $\hat{X}$ to each PC and how it changes with learning can be understood entirely by the growth of the singular values of $W$.

Having linked sensitivity to the weight matrix $W$, all that remains is to show how the input statistics affect the growth of the singular values of the weight matrix. The result from the theory of gradient descent learning in linear networks says that each $\sigma_i$ grows at a different rate. Specifically, for this objective of reconstruction, the growth rate of $\sigma_i$ is proportional to the standard deviation of the corresponding $i$th PC in the data (see Methods). These standard deviations decay as a power law for natural images and are shown in the spectrum in Fig. 5b. As a result, the network output will become sensitive first to the first (largest-variance) PCs and later to the later PCs. This is verified empirically in Fig. 5c. Only at infinite training times does the weight matrix encode all PCs equally and recover the exact solution $W = I$. At any finite learning time, the output of the linear network will be more sensitive to the earlier PCs. Note that this non-uniformity in sensitivity emerges despite the lack of any constraints on $W$ other than learning.

Having introduced this model to explain our findings in artificial networks, we next wondered how it would compare to human behavioral data. We first examined the sensitivity of the linear model to the spatial frequency of a sinusoidal grating (Fig. 5e). In adult humans, the detection threshold to changes in frequency increases linearly with frequency (Fig. 5g)[44]. To compare to human data, we can plot the "detection threshold" of an artificial network as the inverse squared sensitivity of the network output to frequency. This is proportional to the error rate of an optimal read-out of frequency given injected Gaussian noise[45]. At several snapshots during training, we observed that the spatial frequency threshold increased linearly with frequency above a certain cutoff frequency, below which the threshold saturated at a low value (Fig. 5f). Remarkably, even a single matrix trained to

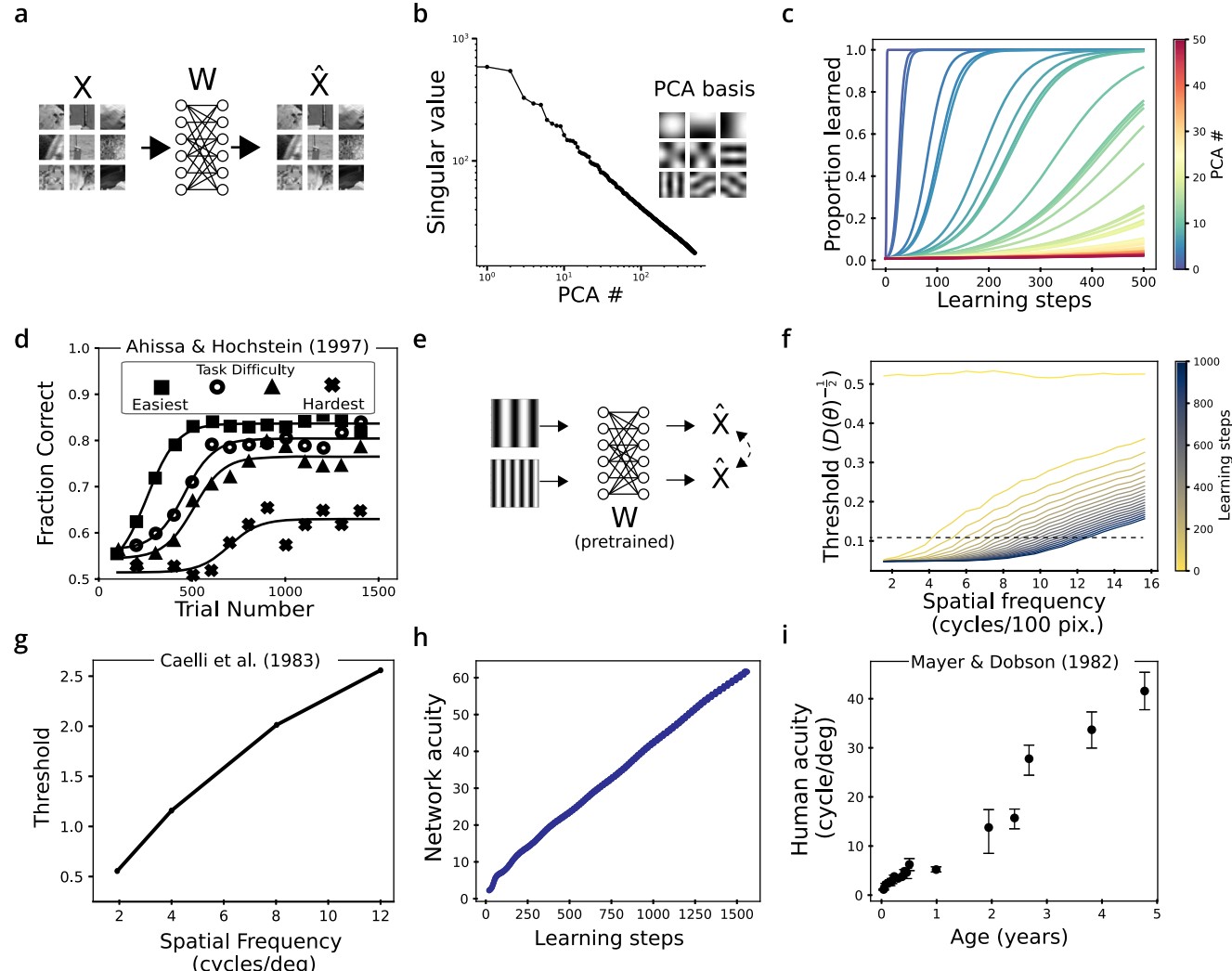

**Fig. 5 | The effect of input statistics on network sensitivity can be understood with linear network models.** Despite their simplicity, these show human-like learning phenomena. **a** We trained linear networks to reconstruct black and white patches of natural images. **b** The statistics of natural images can be analyzed with Principal Components Analysis (PCA); the variance of each successive PC decreases with a characteristic power law decay. **c** When learning with gradient descent, the weight matrix W learns each PC separately and in order of their variance. The sharpness of the sigmoidal learning curve is controlled by the network depth (SI Fig. 2) **d** Human perceptual learning curves are also sigmoidal, and increasing task difficulty delays learning dynamics. Data replotted from ref. 49; subjects trained to detect the orientation of a line, and the difficulty of the task was controlled by a masking stimulus. **e–i** Paradigm for measuring the sensitivity to spatial frequency of W. **f** Every 50 learning steps we plotted the inverse square root of the sensitivity to spatial frequency, which is a proxy for detection thresholds. At each step note the linear increase above an elbow frequency. **g** Human data on spatial frequency thresholds, replotted from ref. 44. **h** An artificial spatial 'acuity' grows nearly linearly with training; 'acuity' is defined as the maximum spatial frequency for which the artificial threshold is below a value of 0.1. **i** In infants and children, the spatial acuity - the highest spatial frequency observable for high-contrast gratings - also increases linearly with age. Replotted from ref. 12, with error bars representing ± SEM over $n$ = 4–10 subjects (varying per point, exact number not reported).

reconstruct images reproduces human-like sensitivity to spatial frequency when trained with gradient descent.

If the human perceptual system is also implicitly constrained by learning dynamics, this would be apparent in psychophysical studies of young children. Indeed, the highest observable frequency of a sinusoidal grating continues to improve with age even up to adolescence (Fig. 5i)[12,46]. This is experience-dependent; when sight is restored in young children by the removal of cataracts their spatial acuity gradually improves[15]. These effects can be reproduced in our model of linear image reconstructions. Defining the network's spatial acuity as the highest spatial frequency whose simulated detection threshold (inverse squared sensitivity) was below a fixed cutoff, we found we could reproduce a linear increase of spatial acuity with experience (Fig. 5h). Learning with gradient descent reproduces not only an efficient encoding of spatial

frequency but also the way in which visual acuity increases linearly with age.

The theory of learning in deep networks makes several further predictions for human perceptual learning, many of which have been explored previously[47,48]. A central feature of this framework is a characteristic sigmoidal curve for perceptual learning tasks (Fig. 5c). Such sigmoidal learning curves are observable in humans on perceptual learning tasks that are sufficiently difficult (Fig. 5d)[49]. This curve is sigmoidal because the rate of improvement depends upon the current level of sensitivity as well as the difference from asymptotic sensitivity (see Methods). This causes sensitivity to rise exponentially at first but eventually converge exponentially towards an asymptote. In human perceptual learning experiments, the learning curve is indeed better described as an exponential than other functional forms such as power laws[50]. Although gradient descent in linear systems is a simple model, it

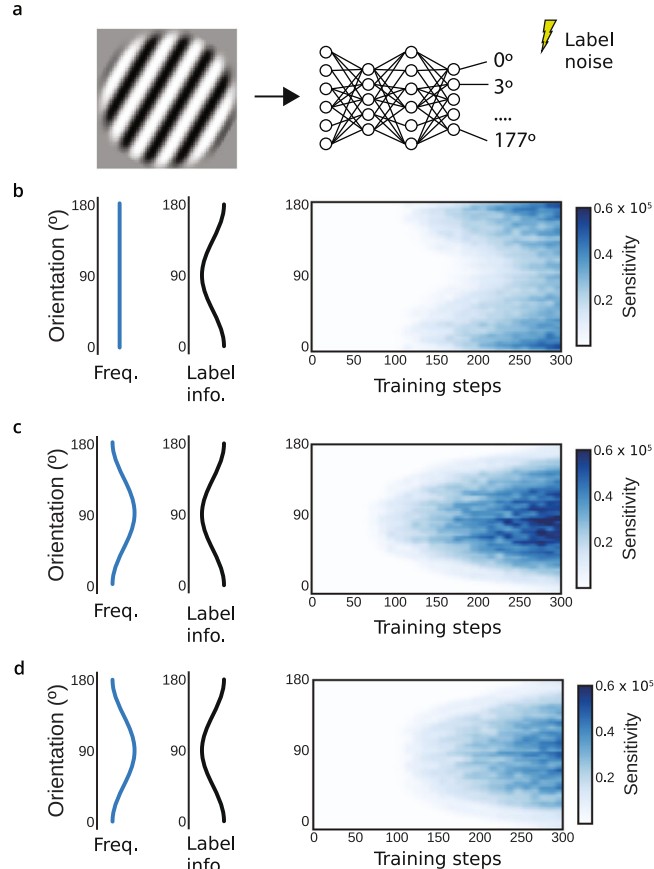

**Fig. 6 | Dissociating the effect of frequency and information in supervised learning tasks. a** We trained 3-layer nonlinear neural networks to classify the orientation of sinusoidal gratings into 3° bins, varying either input frequency or output noise. **b** We controlled the informativeness of input orientations by injecting noise into the labels as a function of orientation (specifically, the binary labels are multiplied by a Bernoulli dropout with a rate that depends on orientation). The sensitivity of the first layer to input orientation is shown over learning. With uniform statistics, the more informative features are preferentially learned. **c** The effect of varying input frequency without applying label noise. In this case, the more frequent features are preferentially learned. **d** We then balanced noise and frequency such that the total information in the input dataset about each output label is uniform (see Methods). Learning with gradient descent still prefers common angles.

accurately captures the functional form of how perception improves with experience.

It is important to note the mathematical reason why gradient descent learns frequent inputs first also applies to supervised learning. The learning rate for a feature is proportional to *both* its correlation with the outputs and to its input variance. Thus, for two features with equal correlation with the output labels but different variance in the inputs, the network will learn to use the higher-variance feature first (see Methods for derivation). Note that if a feature is orthogonal to the task, it is still not learned by the network. Due to this additional pressure on feature learning beyond task usefulness, networks trained on a wide range of objectives will show greater sensitivity for frequent features, as long as this differential pressure is not outweighed by a greater differential in task usefulness. Thus, if finite-learning and implicit biases indeed underlie human perceptual sensitivity patterns as we suggest, the objective function learning of perception may be different than efficient coding per se.

The emergence of efficient coding in supervised tasks can be verified with a simple task in which the frequency and usefulness of input features are varied independently (Fig. 6). This task also

demonstrates that the frequency-dependence of sensitivity is independent from variations in the utility for solving the task. We trained a nonlinear 3-layer neural network to decode the orientation of a sinusoidal grating appearing with a set probability distribution. We also applied noise to the output labels to control the information in each stimulus about the labels. As expected, both the input frequency and output noise separately affect the sensitivity of learned representations (Fig. 6b,c). To demonstrate that gradient descent introduces an additional bias beyond task usefulness, defined as the total information in the input dataset about each label, we next adjusted the magnitude of the noise such that the total information is uniform across labels. This requires applying a greater level of noise onto the labels that are more common, balancing their effects on information. Even in this case, a higher sensitivity to input orientation emerges for more common orientations (Fig. 6d). Note that label information is still crucial; if a feature has no information about labels, it is not learned even if it is frequent. This now provides a deeper intuition of our findings earlier that networks trained on object recognition are more sensitive to frequent features. The preference for frequent features is a general feature of learning with gradient descent and is separate from frequency's effect on the information about labels.

## Discussion

Here we found that the internal representations of artificial neural networks trained on ImageNet are more sensitive to basic visual features that are more common, which is a hallmark feature of efficient coding. We show that this hallmark naturally emerges from gradient-based learning. Even a minimal model of gradient-based learning - linear image reconstruction - reproduced human patterns of sensitivity to sensory variables and how these change over development. In this minimal model the dynamics of learning can be understood analytically. The correspondence of sensitivity and statistics emerges due to an implicit bias of gradient descent for common, high-variance aspects of the input data.

Our result provides a proof of principle that patterns of perceptual sensitivity in animals could be explained by a similar phenomenon. If plasticity in the brain approximates the gradient of some task, whether by reinforced Hebbian rules or some other algorithm, neural populations will preferentially encode the strongest dimensions in their inputs. Note that this is an alternative, or perhaps complementary, mechanism for efficient coding as compared to the many local and unsupervised objectives that have previously been considered as hypotheses[51-61]. Any general algorithm approximating gradient descent may produce similar codes when learning towards many objectives.

It is important to note that our mathematical analysis of linear networks is highly simplified and may not accurately describe how learning affects sensitivity in general. A number of considerations complicate a generalization to nonlinear artificial neural networks, let alone brains. Nonlinearity makes linear decompositions inaccurate, and as a result we cannot say the precise features that gradient descent prefers to learn before others in nonlinear networks. New techniques from this emerging field may soon allow a more complete characterization of the dynamics of learning (e.g.[62]). However, despite these caveats, we find that this model is useful to explain why efficient codes emerge in nonlinear artificial networks. It is remarkable that such a simple model of learning also captures qualitative features of human perceptual learning, as well. Learning in linear systems provides a valuable source of intuition for the effects of learning in more complex systems.

While we have shown one mechanism for how learning can induce a statistics/sensitivity correspondence, it is not the only mechanism by which it could do so. Theories of deep learning often distinguish between the "rich" (feature learning) and "lazy" (kernel) regimes possible in network learning[63]. Our models reside in the rich regime, which

involves learning new intermediate representations and assumes a small weight initialization. In the alternative, lazy regime, intermediate representations change little over learning and only a readout function is learned[20]. Interestingly, networks in the "lazy" (kernel) regime evolve under gradient descent as if they were linear in their parameters[64], and furthermore have the inductive bias of successively fitting higher modes of the input/output function as more data is presented[65,66]. The modes are defined differently, however, via the kernel similarity matrix rather than the direct input covariance. An additional potential source of efficient codes is stochasticity present in training that are not present during evaluation. Dropout is one such potential source[67], but of the networks analyzed only the VGG16 model was trained with dropout. Alternatively, it has been proposed that the stochasticity due to selecting training data in batches provides an implicit constraint, despite it not affecting processing itself[22,23,68,69] but see refs. 70, 71. It would be interesting to investigate the extent to which batch size affects an anisotropy of sensitivity. These possibilities are complementary to our linear network analysis, and suggest that a statistics/sensitivity correspondence could be derived for other network regimes.

Our broadest finding – that task-oriented learning can be a mechanism of producing efficient codes – is relevant to the discussion in psychophysics of the nature of the constraints implied by perceptual sensitivity patterns. It has long been recognized that these patterns imply some limitation upon coding capacity. Here, we make the distinction between implicit limitations due to (a lack of) learning and explicit limitations upon the maximum achievable code quality after learning, such as noise, metabolism, or a limited number of neurons. Although these categories limit perception with different mechanisms, they produce similar patterns of perceptual sensitivity. However, they have distinct implications for what happens during perceptual learning. Previously, perceptual improvements during development have been interpreted as a reduction in internal noise accompanied by a continuous maintenance of optimally efficient codes[72,73]. In our framework, there is no need to invoke a reduction in noise to explain improvements since learning naturally creates codes that reflect environmental statistics at all stages of learning. Supporting this viewpoint is evidence that perceptual learning involves increases in the signal-to-noise ratio through neuronal changes that enhance the signal strength[17,18]. To be sure, the nervous system is indeed constrained by hard ceilings such as noise and metabolism. The implicit constraints due to learning are complementary to these and their relative contribution decreases with age and experience. However, note that the learning rate of some attributes may slow because other attributes have been learned sufficiently well to complete the task[74]. Thus, even to the controversial extent that adults have ceased perceptual learning, their non-uniformity of sensitivity might still be due to finite-learning effects.

On which timescales of learning in humans is this framework most relevant? The most direct comparison is likely to be learning on developmental timescales. However, given that the modeled learning curves are also similar those observed in humans on the timescale of hours and days (Fig. 5d[49]), it is also possible that such principles are active on that timescale. On yet shorter timescales, it may also be of interest to examine the dynamics of perceptual adaption. It has been argued that sensory adaptation is a form of efficient coding, optimally re-allocating sensory encoding resources according to recent stimulus history[75], although note that whether improvements in sensitivity accompany adaptation is controversial[76–78]. Thus sensory adaptation and its dynamics might also be explained and predicted by the global objective of a task-dependent learning rule (gradient descent) in a continually updating (i.e., adapting) sensory processing system such as the brain.

Several further predictions can be made using the model system of gradient descent in linear systems. In this model, the rate of perceptual learning can be quantitatively modeled as a function of input statistics, importance, and current performance. These predictions could be verified in experiments that separably vary label noise and input statistics in supervised perceptual learning problems. Additionally, learning in the rich domain predicts that the learning system should represent the outside world in a low-dimensional way, with additional dimensions being added over time according to their variance and importance. As such, these learning dynamics naturally give rise to low-dimensional neural representations[79]. Such learning dynamics may thus underlie the popular idea in neuroscience of low-dimensional neural manifolds (see ref. 80).

A learning framework for perception points to a different sort of normative analysis of why we perceive the way that we do. Optimality can be defined in two ways. It can characterize the maximum achievable code quality, in an information-theoretic sense, given some number of neurons and their biological limitations. Alternatively, one might also describe responses that are optimal given the limited experience by which to learn the statistics of the world. Even ideal observers must learn from limited data, and successful learning from limited data must be constrained[81]. Appropriate learning constraints would be selected for by evolution. Future research may help to unravel these optimal learning algorithms and characterize their sensory consequences.

## Methods
### Stimuli and calculation of network sensitivity
In all networks, we defined the sensitivity of a particular layer to a sensory variable as the squared magnitude of the gradient. For a layer with N nodes and vector of activations **y**, the sensitivity with respect to a sensory variable $\theta$ is:

$$D(\mathbf{y}; \theta) = \sum_i^N \frac{\partial y_i}{\partial \theta}^2 . \tag{1}$$

This definition of sensitivity can be related to the Fisher Information about a sensory variable $\theta$ in an artificial stimuli set. This is relevant for comparisons to human psychophysical data as the notion of sensitivity inferred from discrimination thresholds is the Fisher Information. In particular, our definition of sensitivity can be interpreted as the Fisher Information of systems with internal Gaussian noise of unit variance, and furthermore for the orientation of stimuli within an artificial stimulus ensemble with one stimulus per value of $\theta$. A derivation of this connection can be found in the Mathematical Supplementary Methods.

The sensitivity can be calculated through backpropagation or by the method of finite differences. We created differentiable generators of stimuli in the automatic differentiation framework of Pytorch. This allowed calculating the sensitivity directly via in-built backpropagation methods.

For the figures in the text, we used Gabor stimuli with a spatial frequency of 2 cycles per 100 pixels, a contrast so that pixels span the range of [-1,1] in intensity in units of z-scored ImageNet image intensities, and a Gaussian envelope with $\sigma = 50$ pixels. We marginalized over the phase of the Gabor by averaging the sensitivity calculated with 10 linearly spaced spatial phases tiling the interval $[-\pi, \pi]$. The sinusoidal stimuli input to the linear network varied in spatial frequency, and we similarly averaged sensitivity over spatial phase. Finally, for the hue stimuli, we generated images of a uniform color in HSV color space and converted pixel values to RGB. Results were marginalized over the S and V axes in the range [0.5, 1] which corresponds to the calculation of hue histograms on ImageNet (see below), which necessarily involves binning S and V.

## Deep nonlinear network experiments

To measure the sensitivity of pretrained networks, we first downloaded pretrained ResNet18 and VGG16 (with batch normalization) networks from the Torchvision python package (v0.11) distributed with Pytorch. For the vision transformer, we used a distribution in Python available at https://github.com/lukemelas/PyTorch-Pretrained-ViT. For each layer in these networks, we calculated sensitivity to orientation and hue with the stimuli generators described above. The 'layers' are defined differently for each network. For ResNet, layers are what in this architecture are called residual blocks (each of which contain multiple linear-nonlinear operations). For VGG, layers are the activations following each linear or pooling layer. Layers within the vision transformer are what are called transformer blocks.

We implemented a number of controls to determine the extent to which the observed patterns of sensitivity related to image statistics. We first ran the sensitivity analysis on untrained networks; we used the Pytorch default initialization. To ensure that the architectures do not show inherent patterns only in a certain regime of weight sizes, we calculated sensitivity on a copy of the networks in which the weights were shuffled. We wanted to preserve weight sizes in a layer-specific manner, and so shuffled the weights only within each tensor.

As further control on the effect of image statistics we retrained certain models on a version of ImageNet in which all images were rotated by 45°, or as well in which the hue of images were rotated by 90°. The transformer model was not retrained due to its expense. Image modifications were performed with Torchvision's in-built rotation and hue adjusting image transformations.

We trained all networks using a training procedure identical to that used in official distributions. The algorithm used was stochastic gradient descent with an initial learning rate of 0.1, decaying by a factor of 10 every 30 epochs, as well as a momentum value of 0.9, ridge regularization ('weight decay') of size $10^{-4}$, and a batch size of 256 images. The networks were trained for 90 epochs. To match the original training setup, we augmented the image dataset with random horizontal reflections and random crops of a size reduction factor varying from 0.08 to 1. Note that the random horizontal reflections change the statistics of orientations so as to be symmetric around the vertical axis. After training, the sensitivity was calculated as above.

## ImageNet hue statistics

We wrote a custom script to extract the hue histogram of all pixels in all images in the ImageNet training set. We binned hues with a resolution of 1°, and binned hues over the S and V range [0.5, 1] to focus on strongly colored pixels. The exact range is arbitrary, but importantly matches the range used when calculating network sensitivity.

## Linear network experiments

We first constructed a database of 32x32 images of natural scene image portions. These image portions were extracted from ImageNet[32], made greyscale, and cropped to size. Our constructed dataset contained over 100,000 examples of image portions. We then performed PCA on this dataset using the PCA method in Scikit-Learn[82], and displayed the singular values of the top 1,000 components in Fig. 5.

Our task consisted of reconstructing these image portions using a single- or multilayer fully-connected linear neural network. To ensure no architectural bottleneck exists, the internal (hidden) dimension of the multilayer network remained at $32^2$, the same as the input and output. The initial parameter values of the networks were scaled down by a factor of 100 from the default Pytorch initialization to ensure rich-regime learning. Networks were trained to minimize the mean-squared error of reconstruction using stochastic gradient descent, a learning rate of 1.0, and a batch size of 16,384, the largest that would fit in memory. The large batch size minimizes effects relating to batch stochasticity.

During learning, we calculated the sensitivity to spatial frequency as well as the projection of the learned weight matrix upon the PCA basis vectors of the inputs. The projection upon each PCA vector is given by $u_i^T W u_i$, where $W$ is the product matrix corresponding to the linear network and $u_i$ is the $i$th PCA component. The sensitivity to spatial frequency was calculated by constructing a sinusoidal plane wave test stimulus with parameterized frequency and phase and using Pytorch's automatic differentiation capability to obtain the derivative of network output with respect to frequency. The sensitivity was calculated for 64 equally-spaced phase offsets and the result averaged over phase.

## Supervised label noise experiment (Fig. 6)

In this experiment we trained a 3-layer neural network with ReLU nonlinearities to decode the orientation of 64x64 pixel image of a sinusoidal grating. The period of the sinusoid was 12.8 pixels, and in each stimulus the sinusoid carried a random phase offset. The random phase and orientation ensured that no image was repeated. In each image the orientation was sampled in the interval $[0, \pi]$ from a specified probability distribution (either a uniform distribution or $\frac{2-\cos(2x)}{2\pi}$). The objective was the categorization of images into 60 bins of orientations, with success quantified via a cross-entropy loss function.

The addition of noise to the output labels was calibrated such that, on average over a dataset, any orientation $\theta$ is as informative as any other despite a potentially nonuniform orientation distribution $p(\theta)$. Since the total information in a dataset about a (potentially noised) label $y_\theta$ scales linearly with how often it appears, all else held equal, the variation in per-example information must exactly balance the change in frequency. That is, for any two orientations $\theta_i$ and $\theta_j$ and their corresponding (noised) labels $y_{\theta_i}$ and $y_{\theta_j}$, it must be that $p(\theta_i)I[y_{\theta_i}|\theta_i] = p(\theta_j)I[y_{\theta_j}|\theta_j]$. Here $I[\cdot]$ represents the information gained about a label having observed an input, i.e. the change in entropy over $y_\theta$ from the uniform distribution. This proportionality is satisfied if $I[y_{\theta_j}|\theta_j] = \frac{a}{p(\theta)}$ for some constant $a$.

Our approach thus requires applying label noise of a known entropy that varies with orientation. Because we optimize a cross-entropy objective, rather than e.g. a mean-squared-error objective, there are no interactions between neighboring bins. We applied noise by treating the nonzero element of each label vector, which are indicator (1-hot) vectors, as a Bernoulli variable with rate $\sigma(\theta)$. $\sigma = 1$ corresponds to the zero-noise condition, and with rate $1 - \sigma(\theta)$, a label is dropped out. For this noise, the information about each label is $I[y_\theta|x_\theta] = 1 - H_b(\sigma(\theta))$, where $H_b(\sigma)$ is the binary entropy function. Together, the rate of Bernoulli noise is given by $\sigma(\theta) = H_b^{-1}\left(1 - \frac{a}{p(\theta)}\right)$. We approximated the inverse binary entropy function with a table lookup and assuming $\sigma \geq 0.5$.

## Sensitivity analysis of linear networks

In this section we will analyze the sensitivity of a linear multilayer neural network in which the weights of layer $i$ are parameterized by $W_i$. The output of such a network with N layers is:

$$Y = W_N W_{N-1} \ldots W_2 W_1 X \qquad (2)$$

The product matrix is simply $W$, such that $Y = WX$.

Throughout this analysis we will make heavy use of the singular value decomposition of the weight matrix, which defines matrices $U, S$, and $V$ such that $W = USV^T$. The matrix $S$ is diagonal, and the diagonal elements are called the singular values $\sigma_i$. The $U$ and $V$ matrices are orthonormal.

Our analysis describes how learning dynamics in this system acts to link output sensitivity to input statistics. Note that the derivation here is for arbitrary objectives; the instance of a reconstruction loss discussed in the main text is a special case. The analysis is organized in three stages: 1) how the sensitivity depends on the singular values $\sigma_i$ of

$W$, 2) how $\sigma_i$ change with learning, and 3) how $\sigma_i$ correspond to the image statistics.

## Sensitivity depends on the singular values of $W$

We wish to derive the sensitivity of a linear network to arbitrary input features. We will first examine the case of determining the sensitivity of the network for the following feature: how much the data aligns with each $j$th singular vector of $W$. This is a weight-dependent feature. Specifically, let the feature $\theta_j$ be the dot product of the data with the $j$th right singular vector of the weight product matrix, $\theta_j = V_j^T x$. This feature is important as it can be used to analyze the sensitivity to arbitrary features.

For this feature, we find the sensitivity of the network is $D(Y; \theta_j) = \sigma_j^2$. This result is intuitive, as $\sigma_j^2$ describes how much data lying along the vector $v_j$ is amplified when multiplied with $W$. A derivation can be found in the Supplementary Methods. Thus, when $\theta_j = V_j^T \mathbf{x}$, the sensitivity $D(Y; \theta_j)$ is constant and is the square of the associated singular value.

The sensitivity to more general $\theta$ can be understood using this result. This is because the key derivative can be decomposed into the derivatives with respect to right singular vectors: $\frac{\partial y_\mu}{\partial \theta} = \frac{\partial y_\mu}{\partial V^T \mathbf{x}}^T \frac{\partial V^T \mathbf{x}}{\partial \theta}$. In this case, we find that $D(y; \theta) = \sum_j \sigma_j^2 \frac{\partial V_j^T \mathbf{x}}{\partial \theta}^2$ (see Supplementary Methods for derivation). Thus, for arbitrary $\theta$, the sensitivity depends on how $V_j^T \mathbf{x}$ depends on $\theta$ times the size of the associated singular value, summed over components $j$.

## The behavior of the singular values

Previous literature describes how the weight matrix changes due to gradient descent[24,28]. More information about these results can be found in the Supplementary Methods.

We first define a (potentially data-dependent) cost function:

$$\ell(W_N W_{N-1} \ldots W_2 W_1) \qquad (3)$$

As described by ref. 24, under certain conditions on the weight initialization the direction of the unit vectors $\mathbf{u}$ and $\mathbf{v}$ rotate with learning in a specific way. Note that they remain unit length during learning. This result, quoted in the Mathematical Supplementary Methods as Theorem 2, states that the vectors are static when they align with the singular vectors of the gradient of the loss, $\nabla \ell(W)$. More specifically, if the singular vectors are static then $U^T \nabla \ell(W) V$ is diagonal. This will become an important condition for tying the input statistics to the singular values of $W$.

Another important result from previous literature describes how singular values of the product matrix $W$ evolve as a function of time $t$:

$$\dot{\sigma}_i(t) = -N \sigma_i(t)^{\frac{2(N-1)}{N}} \langle \nabla_W \ell(W(t)), \mathbf{u}_i(t) \mathbf{v}_i^T(t) \rangle \qquad (4)$$

$$= -N \sigma_i(t)^{\frac{2(N-1)}{N}} \mathbf{u}_i^T(t) \nabla_W \ell(W(t)) \mathbf{v}_i(t) \qquad (5)$$

Thus, each singular value evolves as a product of a function its current size and the network depth ($N \sigma_i(t)^{\frac{2(N-1)}{N}}$) multiplied by how much the gradient correlates with the corresponding singular vectors. This formalism assumes continuous learning dynamics; see[25] for a treatment of finite step sizes.

## Relation of frequency to sensitivity

In this section we wish to show how the input statistics affect the singular vectors and values of $W$. Our approach is to show that frequency $p(\theta)$ reflects in the covariance of $\theta$. The covariance affects the rate of learning of the singular values of the weight matrix $W$.

## Frequency vs. variance

In our analysis of how the statistics of data affect sensitivity, we focus on the variance of features. Since previous literature in psychophysics focuses on frequency as defined by the vector $p(\theta)$ with a scalar value for each orientation $p(\theta = \theta_j)$ (e.g.[8]), it is appropriate to discuss their relation. Our analysis focuses on variance in part because attributes like orientation can occur with a real-valued strength in each image patch when measured by e.g. Gabor filters or Fourier decomposition. Thus a description of $p(\theta)$ in natural images might be more completely characterized with a two-dimensional matrix with dimensions for angle and intensity. Variance summarizes the intensity axis and characterizes how unpredictable each orientation is within each image patch. The second reason we work with variance is that it cleanly relates to the speed of learning.

When features are binary and either present or not, variance and frequency are closely related. Modeling presence as a Bernoulli variable, the frequency is the probability $p(\theta_j)$ and the variance is $\sigma(\theta_j) = p(\theta_j)(1 - p(\theta_j))$. Note that at very small values of $p(\theta_j)$, $\sigma(\theta_j) \sim p(\theta_j)$. However, features that are nearly always present ($p(\theta_j)$ near 1) have a very low variance. It is interesting to note that this behavior aligns with the expectation that efficient sensory systems should dedicate more resources to features whose presence is uncertain ($p(\theta_j) = 0.5$) than to those whose presence is guaranteed ($p(\theta_j) = 1$). Variance is thus very similar to absolute frequency for rare Bernoulli variables and in general may be a more intuitive measure of feature importance in regards to what determines efficient patterns of sensitivity.

## What $W$ learns: autoencoding objective

Further describing the growth of singular values requires a choice of objective. The base case of our study is the autoencoding objective defined for a set of inputs $X$:

$$\ell(W) = \frac{1}{N} \sum_i^N (\mathbf{x}_i - W\mathbf{x}_i)^T (\mathbf{x}_i - W\mathbf{x}_i) \qquad (6)$$

Our goal is to determine how $W$ evolves for this cost function. We will examine both the singular vectors and the singular values.

During learning, the singular vectors rotate (recall they are unit length and orthogonal) until they reach a fixed point. For this cost function, it is easy to verify that a fixed point of dynamics is when the singular vectors are equal to the principal components of the inputs (see Supplementary Methods for proof). That is, the vectors are static when $\Sigma_{xx} = V\Lambda V^T$ and $W = VSV^T$ for the same $V$ but potentially different $\Lambda$ and $S$. This alignment is especially relevant given the expression for network sensitivity derived above. With the vectors aligned, the sensitivity to each corresponding principal component of the inputs is given by $\sigma_i^2$, the squared singular value of $W$.

The evolution of sensitivity is thus governed by the evolution of singular values. The rate of change of $\sigma_i$ is complicated to calculate because the singular vectors can potentially rotate. However, for the sake of analysis one can examine the case when the singular vectors are initialized at the fixed point mentioned above, as in previous literature[28]. In this set of initial conditions, the time-evolution of each singular value of W is given by refs. 24, 28:

$$\dot{\sigma}_i(t) = N\lambda_i \sigma_i(t)^{\frac{2(N-1)}{N}} (1 - \sigma_i(t)) \qquad (7)$$

Note that the rate of learning is controlled by $\lambda_i$, the standard deviation of the $i$th principal component of the inputs. The term on the right causes $\sigma_i(t)$ to converge to 1 asymptotically, as is expected as the solution of the full-rank reconstruction problem is $W = I$. For deeper networks ($N \geq 2$), the growth is sigmoidal and approaches a step function as $N \to \infty$ (see ref. 25). Thus, in this axis-aligned initialization, the singular values $\sigma_i(t)$ are learned in order of the variance of the associated principal components of the inputs.

Together, these results mean that the sensitivity of a linear network's output to the principal components of the inputs evolve in order of variance when trained on input reconstruction. This is exactly the case for the axis-aligned initialization and approximately true for small initializations. For the single-matrix network displayed in the figure in the main text, the sensitivity to the $j$th PC thus evolves over time $t$ as:

$$\dot{D}(Y; PC_j)(t) = 2\lambda_j \sqrt{D(Y; PC_j)(t)}\left(1 - \sqrt{D(Y; PC_j)(t)}\right) \quad (8)$$

### What W learns: supervised learning

We can also determine how input statistics affect the sensitivity for the more general class of objective functions when $Wx$ is trained to match some target $y$ by minimizing the mean-squared error:

$$\ell(W) = \sum_j (\mathbf{y}_j - W\mathbf{x}_j)^2 \quad (9)$$

As before, we can gain intuition about W by beginning from an initialization that is axis-aligned with the final solution. For the supervised case, these initializations share the singular vectors of the data/labels, but can differ in the singular values. Given $\Sigma_{xx} = V\Lambda V^T$ and $\Sigma_{xy} = UTV^T$, we set $W(0) = USV^T$ for the same $U$ and $V$. See the Supplementary Methods for proof that this is a fixed point of singular vector dynamics.

This initialization allows us to understand how the singular values of the weight matrix change. As derived in the Supplementary Methods, the time evolution of $\sigma_i$ is given by:

$$\dot{\sigma}_i(t) = -N\sigma_i(t)^{\frac{2(N-1)}{N}}(\sigma_i(t)\lambda_i - t_i) \quad (10)$$

$$= \lambda_i N\sigma_i(t)^{\frac{2N-1}{N}}\left(\frac{t_i}{\lambda_i} - \sigma_i(t)\right) \quad (11)$$

As in the case for input reconstruction, the $i$th singular value approaches a target. Instead of 1, this value is $\frac{t_i}{\lambda_i}$, the ratio of the importance of this component (the input/output singular value $t_i$) and the standard deviation of that component in the inputs $\lambda_i$. The growth rate is controlled by the distance from this asymptote (right term) and as well as on the input statistics $\lambda_i$. Thus, even for the case of supervised learning the input statistics affect what is learned first via gradient descent directly through $\Sigma_{xx}$ via $\lambda_i$, and not just through the input/label covariance $\Sigma_{xy}$.

### Statistics and reproducibility

All neural network experiments are somewhat stochastic in their results due to a random initialization and a random subselection of training data used in any network update step. Nevertheless the figures are shown with a single network, due in part to the expense of training multiple networks on ImageNet and because we did not compute statistics of network characteristics. No data were excluded from the analyses, and the Investigators were not blinded to outcome assessment.

### Reporting summary

Further information on research design is available in the Nature Portfolio Reporting Summary linked to this article.

## Data availability

All image datasets associated with this study are available as distributed with Torchvision v0.10.0.

## Code availability

All code used to create the figures is available at https://github.com/KordingLab/ANN_psychophysics. This code is citable separately (https://doi.org/10.5281/zenodo.7387922).

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

## Acknowledgements
The authors are appreciative to research assistance in earlier versions of this project from Ryan Guan and Ryan Jeong.

## Author contributions
A.B. wrote the manuscript, prepared the formal analysis, and prepared figures. A.B., L.Z., and C.Q., prepared software. All authors contributed to conceptualization and edited the manuscript. K.K. and A.S. supervised and funded the project.

## Competing interests
The authors declare no competing interests.
