## [Peer Review File · Nature Communications]

Efficient neural codes naturally emerge through gradient descent learningEditorial Note: This manuscript has been previously reviewed at another journal that is not operating a transparent peer review scheme. This document only contains reviewer comments and rebuttal letters for versions considered at *Nature Communications*.

REVIEWERS' COMMENTS

Reviewer #1 (Remarks to the Author):

I thank the authors for the clarifications, and the improvements to an already pretty good manuscript.

However, I would like to push back slightly on a couple of points:

- the fact that DNN regularization does not constitute "resource constraints" by some metrics is fine, as long as the criteria for defining a DNN as "not resource constrained" are explicitly provided in the text as a justification of the term. In the computational neuroscience realm weight regularization is perhaps THE most common way of enforcing resource constraints, together with regularizing the activations, so your definition would not be intuitive to that part of your audience.

- how consistent the reported effects are across a range of task-specific losses is hard to say and thus I still believe that the tone and emphasis in the presentation of the results with respect to the task constraints is too strong for what is actually being presented.

- while replies were provided to my questions in the rebuttal, it wasn't clear to me when anything had changed in the actual manuscript as a way of clarifying the explanation and addressing each of the issues in the text. From what I can tell, beyond adding a couple of citations little changed in the text in light of my comments.

typos and co:

line 67 "resources resources"

Reviewer #2 (Remarks to the Author):

The authors have fully addressed/replied to my comments. It seems to me they also addressed well the other reviewers' comments. I believe this is a valuable contribution to neuroscience, as it offers a new way to think about efficient coding, which also explains some aspects of human perceptual learning and development.

Reviewer #3 (Remarks to the Author):

The authors have done an excellent job responding to the reviews. The paper should be published.

REVIEWERS' COMMENTS

Reviewer #1 (Remarks to the Author):

I thank the authors for the clarifications, and the improvements to an already pretty good manuscript.

However, I would like to push back slightly on a couple of points:

- the fact that DNN regularization does not constitute "resource constraints" by some metrics is fine, as long of the criteria for defining a DNN as "not resource constrained" are explicitly provided in the text as a justification of the term. in the computational neuroscience realm weight regularization is perhaps THE most common way of enforcing resource constraints, together with regularizing the activations, so your definition would not be intuitive to that part of your audience.

Thank you. We agree that this is useful. Where we claimed DNNs are unconstrained, such as in line 174 in Results, we now state that we mean by the learning theory metric of generalization. We write they "can encode an extremely large amount of information; for example, they are capable of memorizing millions of examples of random noise (Zhang et al. 2021)."

- how consistent the reported effects are across a range of task-specific losses is hard to say and thus I still believe that the tone and emphasis in the presentation of the results with respect to the task constraints is too strong for what is actually being presented.

To address this, we have added cautionary sentences in the relevant Results paragraph. These emphasize that the task matters as well. Our results show that, regardless of task, there is a *pressure* to learn more frequent features, but this may be outweighed by task usefulness.

The paragraph now reads (additions in bold),

"It is important to note the mathematical reason why gradient descent learns frequent inputs first also applies to supervised learning. **The learning rate for a given input feature is proportional to both its correlation with the outputs and its input variance.** Thus, for two features with equal correlation with the output labels but different variance in the inputs, the network will learn to use the higher-variance feature first (see Methods for derivation). **Note that if a feature is orthogonal to the task, it is still not learned by the network.** Due to this additional pressure on feature learning beyond task usefulness, networks trained on a wide range of objectives will show greater sensitivity for frequent features, **as long as this differential pressure is not outweighed by a greater differential in task usefulness.** Thus, if finite-learning and implicit biases indeed underlie human perceptual sensitivity patterns as we suggest, the objective function learning of perception may be different than efficient coding *per se.*"

- while replies were provided to my questions in the rebuttal, it wasn't clear to me when

anything had changed in the actual manuscript as a way of clarifying the explanation and addressing each of the issues in the text. From what i can tell, beyond adding a couple of citations little changed in the text in light of my comments.

We apologize; below, we've summarized the substantive changes we had made in response to the original review.

- In response to the question about the location and size of the input Gabor, we ran a new analysis with varying sizes and locations, which is now included as supplemental figure 3.
- We also ran a new analysis in response to the question about using the Adam optimizer, rather than gradient descent. This now appears as supplemental figures 2b and 3.
- As requested, we reporting the specific levels of L2 regularization in Methods (line 490). This was an important detail.
- We rephrased the statement "before training randomly initialized networks largely do not show this pattern " for clarity; the new statement is on line 141.
- In response to the question about the confound of stochasticity, we added several sentences to the discussion, and along with multiple accompanying citations (lines 372-377)
- We changed the verbal definition of gradient descent in the introduction, line 80
- The related phenomenological model of plasticity pointed out by the reviewer, Bredenberg 2020, is now cited in the discussion Line.
- We more carefully phrased our mention of 'unconstrained' throughout, as exemplified in the paragraph reproduced above.

These changes were important and we think improved the manuscript. We are sorry to not have called them out clearly.

typos and co:
line 67 "resources resources"

Reviewer #2 (Remarks to the Author):

The authors have fully addressed/replied to my comments. It seems to me they also addressed well the other reviewers comments. I believe this is a valuable contribution to neuroscience, as it offers a new way to think about efficient coding, which also explains some aspects of human perceptual learning and development.

Reviewer #3 (Remarks to the Author):

The authors have done an excellent job respond to the reviews. The paper should be published.